# Minimax Classification with 0-1 Loss and Performance Guarantees

**Santiago Mazuelas**
BCAM-Basque Center for Applied Mathematics
and IKERBASQUE-Basque Foundation for Science
Bilbao, Spain
smazuelas@bcamath.org

**Andrea Zanoni**
École Polytechnique Fédérale de Lausanne
Lausanne, Switzerland
andrea.zanoni@epfl.ch

**Aritz Pérez**
BCAM-Basque Center for Applied Mathematics
Bilbao, Spain
aperez@bcamath.org

## Abstract

Supervised classification techniques use training samples to find classification rules with small expected 0-1 loss. Conventional methods achieve efficient learning and out-of-sample generalization by minimizing surrogate losses over specific families of rules. This paper presents minimax risk classifiers (MRCs) that do not rely on a choice of surrogate loss and family of rules. MRCs achieve efficient learning and out-of-sample generalization by minimizing worst-case expected 0-1 loss w.r.t. uncertainty sets that are defined by linear constraints and include the true underlying distribution. In addition, MRCs' learning stage provides performance guarantees as lower and upper tight bounds for expected 0-1 loss. We also present MRCs' finite-sample generalization bounds in terms of training size and smallest minimax risk, and show their competitive classification performance w.r.t. state-of-the-art techniques using benchmark datasets.

## 1  Introduction

Supervised classification techniques use training samples to find classification rules that assign labels to instances with small expected 0-1 loss, also referred to as risk or probability of error. Most learning methods utilize empirical risk minimization (ERM) approach that minimizes the expectation w.r.t. the empirical distribution of training samples, see e.g., [1, 2]. Other methods utilize robust risk minimization (RRM) approach that minimizes the worst-case expectation w.r.t. an uncertainty set of distributions obtained using metrics such as moments' fits, divergences, and Wasserstein distances, see e.g., [3, 4]. Common uncertainty sets are formed by distributions with instances' marginal supported on the training samples [5–9]. However, more general uncertainty sets, such as those used in [3, 4, 10–12], can include the true underlying distribution with a tuneable confidence. Out-of-sample generalization is conventionally achieved by considering families of rules with favorable properties (reduced VC dimension or Rademacher complexity [1, 13]). However, RRM techniques can directly achieve out-of-sample generalization by using uncertainty sets that include the true underlying distribution. In addition, such uncertainty sets can enable to obtain tight performance bounds at learning.

Conventional methods achieve efficient learning and out-of-sample generalization by minimizing surrogate losses over families of rules with favorable properties. ERM-based techniques such as support vector machines (SVMs), multilayer perceptrons (MLPs), and Adaboost classifiers consider

loss functions such as hinge loss, cross-entropy loss, and exponential loss together with families of classification rules obtained from reproducing kernel Hilbert spaces (RKHSs), artificial neural networks, and combinations of weak rules. RRM-based techniques that utilize Wasserstein distances consider surrogate log loss and linear functions or RKHSs [11, 12], while those that utilize f-divergences can use more general surrogate losses and parametric families of rules as long as they result in convex functions over parameters [8, 9]. Certain techniques based on RRM do not rely on surrogate losses and minimize worst-case 0-1 expected loss [5–7]. However, such works consider uncertainty sets that do not include the true underlying distribution. Hence, their generalization guarantees rely on the usage of specific families of rules, and they do not provide performance bounds at learning.

This paper presents RRM-based classification techniques referred to as minimax risk classifiers (MRCs) that minimize worst-case expected 0-1 loss over general classification rules, and provide tight performance bounds at learning. Specifically, the main results presented in the paper are as follows.

- Learning techniques that determine MRCs as the solution of a linear optimization problem (Theorem 1 in Section 2, and Algorithm 1 in Section 4).
- Techniques that provide performance guarantees at learning as lower and upper tight bounds for expected 0-1 loss (Theorem 1 in Section 2, Theorem 2 in Section 3, and Algorithm 1 in Section 4).
- Finite-sample generalization bounds for MRCs in terms of training size and smallest minimax risk (Theorem 3 in Section 3).

Detailed comparisons with related techniques are provided in the remarks to the paper's main new results. In addition, Section 4 provides a detailed description of MRCs' implementation, and Section 5 shows the suitability of the performance bounds and compares the classification error of MRCs w.r.t. state-of-the-art techniques.

*Notation:* calligraphic upper case letters denote sets; vectors and matrices are denoted by bold lower and upper case letters, respectively; for a vector $\mathbf{v}$, $v^{(l)}$ denotes its $l$-th component, and $\mathbf{v}^{\mathrm{T}}$ and $\mathbf{v}_+$ denote its transpose and positive part, respectively; probability distributions and classification rules are denoted by upright fonts, e.g., p and h; $\mathbb{E}_{\mathrm{p}}\{\cdot\}$ denotes expectation w.r.t. probability distribution p; $\mathbb{I}\{\cdot\}$ denotes the indicator function; $\preceq$ and $\succeq$ denote vector (component-wise) inequalities; $\mathbf{1}$ denotes a vector with all components equal to 1; $\mathbf{1}_{\mathcal{C}}$ an indicator vector with $j$-th component equal to 1 (resp. 0) if $j \in \mathcal{C}$ (resp. $j \notin \mathcal{C}$); $|\mathcal{Z}|$ denotes de cardinality of set $\mathcal{Z}$; and, for a finite set $\mathcal{Z}$, we denote by $\Delta(\mathcal{Z})$ the set of probability distributions with support $\mathcal{Z}$.

## 2 Minimax-risk classification

This section first briefly recalls the problem statement and learning approaches for supervised classification, and then presents learning techniques for MRCs.

### 2.1 Problem formulation and learning approaches

Supervised classification uses training samples formed by instance-label pairs to determine classification rules that assign labels to instances. In what follows, we denote by $\mathcal{X}$ and $\mathcal{Y}$ the sets of possible instances and labels, respectively; both sets are taken to be finite and we represent $\mathcal{Y}$ by $\{1, 2, \ldots, |\mathcal{Y}|\}$. Commonly, the cardinality of $\mathcal{X}$ is very large compared with that of $\mathcal{Y}$; for instance, in hand-written digit classification with 28x28 pixels grayscale images, $|\mathcal{X}| = 256^{784}$ and $|\mathcal{Y}| = 10$.

Classification rules can be deterministic or non-deterministic. For a specific instance, a deterministic classification rule assigns always the same label, while a non-deterministic classification rule is allowed to randomly assign a label with certain probability. Both types of rules can be represented by the probabilities with which labels are assigned to instances (0 or 1 probabilities for the deterministic case). We denote by $T(\mathcal{X}, \mathcal{Y})$ the set of general classification rules; if $\mathrm{h} \in T(\mathcal{X}, \mathcal{Y})$ we denote by $\mathrm{h}(y|x)$ the probability with which h assigns label $y \in \mathcal{Y}$ to instance $x \in \mathcal{X}$. In addition, we denote by $\Delta(\mathcal{X} \times \mathcal{Y})$ the set of probability distributions on $\mathcal{X} \times \mathcal{Y}$; if $\mathrm{p} \in \Delta(\mathcal{X} \times \mathcal{Y})$ we denote by $\mathrm{p}(x, y)$ the probability assigned by p to the instance-label pair $(x, y)$, and by $\mathrm{p}(x)$ the marginal probability assigned by p to the instance $x$, i.e., $\mathrm{p}(x) = \sum_{y \in \mathcal{Y}} \mathrm{p}(x, y)$.

The 0-1 loss (also called just loss in the following) of a classification rule at the instance-label pair $(x, y) \in \mathcal{X} \times \mathcal{Y}$ quantifies classification error, that is, the loss is 0 if the classification rule assigns label $y$ to instance $x$, and is 1 otherwise. Hence, the expected loss of a classification rule $h \in T(\mathcal{X}, \mathcal{Y})$ at $(x, y)$ is $1 - h(y|x)$, and its expected loss w.r.t. a probability distribution $p \in \Delta(\mathcal{X} \times \mathcal{Y})$ is

$$\ell(h, p) = \sum_{x \in \mathcal{X}, y \in \mathcal{Y}} p(x, y)(1 - h(y|x)).$$

Let $p^*$ be the unknown true underlying distribution of instance-label pairs, the risk of a classification rule $h$ (denoted $R(h)$) is its expected loss w.r.t. $p^*$, that is $R(h) = \ell(h, p^*)$. The minimum risk is known as Bayes risk and becomes

$$R_{\text{Bayes}} = 1 - \sum_{x \in \mathcal{X}} \max_{y \in \mathcal{Y}} p^*(x, y)$$

since it is achieved by Bayes' rule $h_{\text{Bayes}}$ that assigns the most probable label to each instance.

ERM approach for supervised classification aims to minimize the empirical expected loss $\ell(h, p_n)$, where $p_n$ is the empirical distribution of training samples. RRM approach aims to minimize the worst-case expected loss $\ell(h, p)$ for $p$ a probability distribution in an uncertainty set obtained from training samples. As described above, conventional techniques enable efficient ERM and RRM by using surrogate loss functions and considering specific families of classification rules.

Supervised classification techniques can be seen as methods that perform the approximation

$$\min_{h \in T(\mathcal{X}, \mathcal{Y})} \ell(h, p^*) \longrightarrow \min_{h \in \mathcal{F}} \max_{p \in \mathcal{U}} \widetilde{\ell}(h, p)$$

where the original 0-1 loss $\ell$ is substituted by a surrogate loss $\widetilde{\ell}$; classification rules are restricted to a specific family $\mathcal{F} \subseteq T(\mathcal{X}, \mathcal{Y})$; and expectation w.r.t. the true underlying distribution $p^*$ is approximated by the worst-case expectation w.r.t. distributions in an uncertainty set $\mathcal{U}$. ERM-based techniques correspond to the case where the uncertainty set contains only the empirical distribution, while RRM-based techniques use uncertainty sets that contain multiple distributions. Using 0-1 loss and uncertainty sets that include the true underlying distribution, the objective minimized at learning $\max_{p \in \mathcal{U}} \ell(h, p)$ becomes an upper bound of the original objective $\ell(h, p^*)$ for any classification rule $h \in T(\mathcal{X}, \mathcal{Y})$. This key property can enable to ensure out-of-sample generalization and to obtain tight performance bounds at learning.

## 2.2 Learning MRCs

The following shows how RRM can be used with original 0-1 loss $\ell$, considering general classification rules $T(\mathcal{X}, \mathcal{Y})$, and using uncertainty sets that include the true underlying distribution $p^*$ with a tuneable confidence.

MRCs consider uncertainty sets of distributions defined by linear constraints obtained from expectation estimates of a feature mapping. Specifically, let $\Phi : \mathcal{X} \times \mathcal{Y} \to \mathbb{R}^m$ be a feature mapping, and $\mathbf{a}, \mathbf{b} \in \mathbb{R}^m$ with $\mathbf{a} \preceq \mathbf{b}$ be lower and upper endpoints of interval estimates for the expectation of $\Phi$. We consider uncertainty sets of distributions

$$\mathcal{U}^{\mathbf{a}, \mathbf{b}} = \left\{ p \in \Delta(\mathcal{X} \times \mathcal{Y}) : \mathbf{a} \preceq \mathbb{E}_p \{\Phi(x, y)\} \preceq \mathbf{b} \right\} \tag{1}$$

and we denote the minimax expected loss against uncertainty set $\mathcal{U}^{\mathbf{a}, \mathbf{b}}$ by $R^{\mathbf{a}, \mathbf{b}}$, i.e.,

$$R^{\mathbf{a}, \mathbf{b}} = \min_{h \in T(\mathcal{X}, \mathcal{Y})} \max_{p \in \mathcal{U}^{\mathbf{a}, \mathbf{b}}} \ell(h, p).$$

Such uncertainty sets include the true underlying distribution $p^*$ with probability at least $1 - \delta$ as long as $\mathbf{a}$ and $\mathbf{b}$ define expectations' confidence intervals at level $1 - \delta$, that is

$$\mathbb{P}\{\mathbf{a} \preceq \mathbb{E}_{p^*}\{\Phi(x, y)\} \preceq \mathbf{b}\} \geq 1 - \delta.$$

In this paper, we consider expectations' interval estimates obtained from empirical expectations of training samples $(x_1, y_1), (x_2, y_2), \dots, (x_n, y_n)$ as

$$\mathbf{a}_n = \boldsymbol{\tau}_n - \frac{\boldsymbol{\lambda}}{\sqrt{n}}, \ \mathbf{b}_n = \boldsymbol{\tau}_n + \frac{\boldsymbol{\lambda}}{\sqrt{n}}, \ \text{for } \boldsymbol{\tau}_n = \frac{1}{n} \sum_{i=1}^{n} \Phi(x_i, y_i) \tag{2}$$

where $\boldsymbol{\lambda} \succeq \mathbf{0}$ determines the size of the interval estimates for different confidence levels.

In the following, in order to get compact expressions we often denote functions with domain $\mathcal{X} \times \mathcal{Y}$ by vectors or matrices with $|\mathcal{X}||\mathcal{Y}|$ components or rows, respectively. We denote a probability distribution $\mathrm{p} \in \Delta(\mathcal{X} \times \mathcal{Y})$ and a classification rule $\mathrm{h} \in T(\mathcal{X}, \mathcal{Y})$ by vectors $\mathbf{p}$ and $\mathbf{h}$ with components given by $\mathrm{p}(x, y)$ and $\mathrm{h}(y|x)$ for $(x, y) \in \mathcal{X} \times \mathcal{Y}$. In addition, we denote the feature mapping $\Phi : \mathcal{X} \times \mathcal{Y} \to \mathbb{R}^m$ by a matrix $\boldsymbol{\Phi}$ with rows given by $\Phi(x, y)^{\mathrm{T}}$ for $(x, y) \in \mathcal{X} \times \mathcal{Y}$. Also, we denote by $\mathbf{p}_x$, $\mathbf{h}_x$, and $\boldsymbol{\Phi}_x$ the subvectors and submatrix of $\mathbf{p}$, $\mathbf{h}$, and $\boldsymbol{\Phi}$ corresponding to a fixed $x \in \mathcal{X}$, and if $\mathbf{v}$ is a vector indexed by $\mathcal{X} \times \mathcal{Y}$ we denote by $\|\mathbf{v}\|_{1,\infty}$ and $\|\mathbf{v}\|_{\infty,1}$ the mixed norms $\|\mathbf{v}\|_{1,\infty} = \max_{x \in \mathcal{X}} \|\mathbf{v}_x\|_1$ and $\|\mathbf{v}\|_{\infty,1} = \sum_{x \in \mathcal{X}} \|\mathbf{v}_x\|_\infty$. With this vector notation we have that

$$\ell(\mathrm{h}, \mathrm{p}) = \mathbf{p}^{\mathrm{T}}(\mathbf{1} - \mathbf{h}), \quad \min_{\mathrm{h} \in T(\mathcal{X}, \mathcal{Y})} \ell(\mathrm{h}, \mathrm{p}) = 1 - \|\mathbf{p}\|_{\infty,1}, \quad \text{and } \mathbb{E}_{\mathrm{p}}\{\Phi(x, y)\} = \boldsymbol{\Phi}^{\mathrm{T}}\mathbf{p}.$$

Finally, whenever we use expectation point estimates, i.e., $\mathbf{a} = \mathbf{b}$, we drop $\mathbf{b}$ from the superscripts, for instance we denote $\mathcal{U}^{\mathbf{a},\mathbf{b}}$ for $\mathbf{a} = \mathbf{b}$ as $\mathcal{U}^{\mathbf{a}}$.

The result below determines minimax classification rules with 0-1 loss against uncertainty sets given by (1), which are referred to as MRCs in the following.

**Theorem 1.** Let $\Phi : \mathcal{X} \times \mathcal{Y} \to \mathbb{R}^m$, $\mathbf{a}, \mathbf{b} \in \mathbb{R}^m$ with $\mathcal{U}^{\mathbf{a},\mathbf{b}} \neq \emptyset$, and $\boldsymbol{\mu}_a^*, \boldsymbol{\mu}_b^*, \nu^*$ be a solution of the convex optimization problem

$$\min_{\boldsymbol{\mu}_a, \boldsymbol{\mu}_b \in \mathbb{R}^m, \nu \in \mathbb{R}} \quad \begin{aligned} &\mathbf{b}^{\mathrm{T}}\boldsymbol{\mu}_b - \mathbf{a}^{\mathrm{T}}\boldsymbol{\mu}_a - \nu \\ \text{s. t.} \quad &\|(\boldsymbol{\Phi}(\boldsymbol{\mu}_a - \boldsymbol{\mu}_b) + (\nu + 1)\mathbf{1})_+\|_{1,\infty} \leq 1 \\ &\boldsymbol{\mu}_a, \boldsymbol{\mu}_b \succeq \mathbf{0}. \end{aligned} \tag{3}$$

If a classification rule $\mathrm{h}^{\mathbf{a},\mathbf{b}} \in \Delta(X, Y)$ satisfies, for each $x \in \mathcal{X}, y \in \mathcal{Y}$,

$$\mathrm{h}^{\mathbf{a},\mathbf{b}}(y|x) \geq \Phi(x, y)^{\mathrm{T}}\boldsymbol{\mu}^* + \nu^* + 1 \tag{4}$$

with $\boldsymbol{\mu}^* = \boldsymbol{\mu}_a^* - \boldsymbol{\mu}_b^*$, then

$$\mathrm{h}^{\mathbf{a},\mathbf{b}} \in \arg \min_{\mathrm{h} \in \Delta(X, Y)} \max_{\mathrm{p} \in \mathcal{U}^{\mathbf{a},\mathbf{b}}} \ell(\mathrm{h}, \mathrm{p})$$

that is, $\mathrm{h}^{\mathbf{a},\mathbf{b}}$ is a minimax classification rule for 0-1 loss against uncertainty set $\mathcal{U}^{\mathbf{a},\mathbf{b}}$. In addition, the minimax expected loss against uncertainty set $\mathcal{U}^{\mathbf{a},\mathbf{b}}$ is given by

$$R^{\mathbf{a},\mathbf{b}} = \mathbf{b}^{\mathrm{T}}\boldsymbol{\mu}_b^* - \mathbf{a}^{\mathrm{T}}\boldsymbol{\mu}_a^* - \nu^*. \tag{5}$$

*Proof.* See Appendix B in the supplementary material. $\square$

The result above is obtained by using von Neumann's minimax theorem [14] and Lagrange duality [15]; in particular, parameters $\boldsymbol{\mu}_a^*, \boldsymbol{\mu}_b^*, \nu^*$ correspond to the Lagrange multipliers of constraints in (1). As we describe in Section 4, Theorem 1 enables MRCs' implementation in practice. Specifically, training samples serve to obtain expectation estimates $\mathbf{a}$ and $\mathbf{b}$ that are used to learn parameters $\boldsymbol{\mu}^*, \nu^*$ by solving (3), which is equivalent to a linear optimization problem. Then, those parameters are used in the prediction stage to assign label $y \in \mathcal{Y}$ to instance $x \in \mathcal{X}$ with probability $\mathrm{h}^{\mathbf{a},\mathbf{b}}(y|x)$ satisfying (4). Even though MRCs minimize the worst-case risk over all possible rules; as shown in (4), they have a specific parametric form determined by a linear-affine combination of the feature mapping with coefficients obtained by solving (3) at learning. Therefore, the role of the feature mapping in the presented method is similar to that in conventional techniques such as SVM and logistic regression.

Classification rules satisfying (4) always exist since $\sum_{y \in \mathcal{Y}}(\Phi(x, y)^{\mathrm{T}}\boldsymbol{\mu}^* + \nu^* + 1)_+ \leq 1$ for any $x \in \mathcal{X}$ due to the constraints in (3). In addition, in case of using expectation point estimates, i.e., $\mathbf{a} = \mathbf{b}$, the minimization solved at learning becomes

$$\min_{\boldsymbol{\mu} \in \mathbb{R}^m, \nu \in \mathbb{R}} \quad \begin{aligned} &-\mathbf{a}^{\mathrm{T}}\boldsymbol{\mu} - \nu \\ \text{s. t.} \quad &\|(\boldsymbol{\Phi}\boldsymbol{\mu} + (\nu + 1)\mathbf{1})_+\|_{1,\infty} \leq 1 \end{aligned} \tag{6}$$

taking $\boldsymbol{\mu} = \boldsymbol{\mu}_a - \boldsymbol{\mu}_b$.

The techniques proposed in [5–7] find minimax classification rules with 0-1 loss for uncertainty sets that are also defined in terms of expectations' fits. In particular, [6, 7] utilize uncertainty sets of the form

$$\mathcal{U} = \left\{ p \in T(\mathcal{X}, \mathcal{Y}) : \mathbb{E}_p\{\Phi(x, y)\} = \mathbf{a}, \text{ and } p(x) = p_n(x), \ \forall x \in \mathcal{X} \right\}$$

while [5] utilizes uncertainty sets of the form

$$\mathcal{U} = \left\{ p \in T(\mathcal{X}, \mathcal{Y}) : \|\mathbb{E}_p\{\Phi(x, y)\} - \mathbf{a}\| \leq \varepsilon, \text{ and } p(x) = p_n(x), \ \forall x \in \mathcal{X} \right\}.$$

Such uncertainty sets only contain distributions with instances' marginal $p(x)$ that coincides with the empirical $p_n(x)$ so that they do not include the true underlying distribution for finite number of samples. Therefore, the techniques in [5–7] cannot ensure out-of-sample generalization with general classification rules and do not provide performance bounds at learning such as those shown below in Theorem 2 for MRCs.

## 3  Performance guarantees

This section characterizes the out-of-sample performance of MRCs. We first present techniques that provide tight performance bounds at learning, and then we show finite-sample generalization bounds for MRCs' risk in terms of training size and smallest minimax risk.

### 3.1  Tight performance bounds

The following result shows that the proposed approach also allows to obtain bounds for expected losses by solving linear optimization problems.

**Theorem 2.** Let $\Phi : \mathcal{X} \times \mathcal{Y} \to \mathbb{R}^m$, $\mathbf{a}, \mathbf{b} \in \mathbb{R}^m$ with $\mathcal{U}^{\mathbf{a},\mathbf{b}} \neq \emptyset$ and $\kappa^{\mathbf{a},\mathbf{b}}(q)$ be given by

$$
\kappa^{\mathbf{a},\mathbf{b}}(q) \quad = \quad \min_{\boldsymbol{\mu}_a, \boldsymbol{\mu}_b \in \mathbb{R}^m, \nu \in \mathbb{R}} \quad \mathbf{b}^{\mathsf{T}} \boldsymbol{\mu}_b - \mathbf{a}^{\mathsf{T}} \boldsymbol{\mu}_a - \nu
$$
$$
\text{s. t.} \quad \Phi(\boldsymbol{\mu}_a - \boldsymbol{\mu}_b) + \nu \mathbf{1} \preceq \mathbf{q} \qquad (7)
$$
$$
\boldsymbol{\mu}_a, \boldsymbol{\mu}_b \succeq \mathbf{0}
$$

for a function $q : \mathcal{X} \times \mathcal{Y} \to \mathbb{R}$. Then, for any $p \in \mathcal{U}^{\mathbf{a},\mathbf{b}}$ and $h \in T(\mathcal{X}, \mathcal{Y})$

$$0 \leq -\kappa^{\mathbf{a},\mathbf{b}}(1 - h) \leq \ell(h, p) \leq \kappa^{\mathbf{a},\mathbf{b}}(h - 1) \leq 1. \qquad (8)$$

In addition, $\ell(h, p) = -\kappa^{\mathbf{a},\mathbf{b}}(1 - h)$ (resp. $\ell(h, p) = \kappa^{\mathbf{a},\mathbf{b}}(h - 1)$) if $p$ minimizes (resp. maximizes) the expected loss of $h$ over distributions in $\mathcal{U}^{\mathbf{a},\mathbf{b}}$.

*Proof.* See Appendix C in the supplementary material. □

For an MRC $h^{\mathbf{a},\mathbf{b}}$, the upper bound above is directly given by (5), that is, $R^{\mathbf{a},\mathbf{b}} = \kappa^{\mathbf{a},\mathbf{b}}(h^{\mathbf{a},\mathbf{b}} - 1)$. On the other hand, its lower bound, denoted by $L^{\mathbf{a},\mathbf{b}}$, requires to solve an additional linear optimization problem given by (7) to obtain $L^{\mathbf{a},\mathbf{b}} = -\kappa^{\mathbf{a},\mathbf{b}}(1 - h^{\mathbf{a},\mathbf{b}})$.

The techniques proposed in [8, 11, 12] obtain analogous upper and lower bounds corresponding with RRM methods that use uncertainty sets defined in terms of f-divergences and Wasserstein distances. Such methods obtain classification rules by minimizing the upper bound of a surrogate expected loss while MRCs minimize the upper bound of the 0-1 expected loss (risk). Note that the bounds for expected losses become risk's bounds if the uncertainty set includes the true underlying distribution. Such situation can be attained with a tuneable confidence using uncertainty sets defined by Wasserstein distances as in [11, 12] or using the proposed uncertainty sets in (1) with expectation confidence intervals. However, the bounds are only asymptotical risk's bounds using uncertainty sets defined by f-divergences as in [8] or using the proposed uncertainty sets in (1) with expectation point estimates.

### 3.2  Finite-sample generalization bounds

The smallest minimax risk using uncertainty sets given by (1) with feature mapping $\Phi$ is the non-random constant $R^{\boldsymbol{\tau}_\infty}$ with $\boldsymbol{\tau}_\infty = \mathbb{E}_{p^*}\{\Phi\}$ because $p^* \in \mathcal{U}^{\mathbf{a},\mathbf{b}} \Rightarrow \mathcal{U}^{\boldsymbol{\tau}_\infty} \subseteq \mathcal{U}^{\mathbf{a},\mathbf{b}} \Rightarrow R^{\boldsymbol{\tau}_\infty} \leq R^{\mathbf{a},\mathbf{b}}$.

Such smallest minimax risk corresponds with MRC $h^{\tau_\infty}$ that would require an infinite number of training samples to exactly determine the features' actual expectation $\tau_\infty$.

The following result bounds the risk of MRCs w.r.t. the smallest minimax risk, as well as the difference between the risk of MRCs and the corresponding minimax expected loss.

**Theorem 3.** Let $\Phi : \mathcal{X} \times \mathcal{Y} \to \mathbb{R}^m$ be a feature mapping, $\delta \in (0,1)$, and $\tau_\infty = \mathbb{E}_{p^*}\{\Phi\}$. If $\tau_n$, $\mathbf{a}_n$, and $\mathbf{b}_n$ are point and interval estimates for $\tau_\infty$ obtained from training samples as given by (2) with

$$\boldsymbol{\lambda} = \mathbf{d}\sqrt{\frac{\log m + \log \frac{2}{\delta}}{2}}, \ d^{(l)} = \max_{x \in \mathcal{X}, y \in \mathcal{Y}} \Phi(x,y)^{(l)} - \min_{x \in \mathcal{X}, y \in \mathcal{Y}} \Phi(x,y)^{(l)}, \text{ for } l = 1, 2, \ldots, m.$$

Then, with probability at least $1 - \delta$

$$R(h^{\mathbf{a}_n, \mathbf{b}_n}) \leq R^{\mathbf{a}_n, \mathbf{b}_n} \leq R^{\tau_\infty} + 2M_\Phi \|\mathbf{d}\|_2 \sqrt{\frac{\log m + \log \frac{2}{\delta}}{2}} \frac{1}{\sqrt{n}} \tag{9}$$

$$R(h^{\tau_n}) \leq R^{\tau_n} + M_\Phi \|\mathbf{d}\|_2 \sqrt{\frac{\log m + \log \frac{2}{\delta}}{2}} \frac{1}{\sqrt{n}} \tag{10}$$

$$R(h^{\tau_n}) \leq R^{\tau_\infty} + N_\Phi \|\mathbf{d}\|_2 \sqrt{\frac{\log m + \log \frac{2}{\delta}}{2}} \frac{1}{\sqrt{n}} \tag{11}$$

where

$$M_\Phi = \max_{\boldsymbol{\mu} \in \Omega_\Phi} \|\boldsymbol{\mu}\|_2, \ N_\Phi = \max_{\boldsymbol{\mu}_1, \boldsymbol{\mu}_2 \in \Omega_\Phi} \|\boldsymbol{\mu}_1 - \boldsymbol{\mu}_2\|_2$$

$$\Omega_\Phi = \left\{ \boldsymbol{\mu} \in \mathbb{R}^m : \exists \mathbf{a} \in \text{Conv}(\Phi(\mathcal{X} \times \mathcal{Y})) \text{ s.t. } \boldsymbol{\mu}, \nu \text{ is the min. euclidean norm solution of (6)} \right\}.$$

*Proof.* See Appendix D in the supplementary material. □

Second inequality in (9) and inequality (11) bound the risk of MRCs w.r.t. the smallest minimax risk $R^{\tau_\infty}$; and first inequality in (9) and inequality (10) bound the difference between the risk of MRCs and the corresponding minimax expected loss. These bounds show differences that decrease with $n$ as $O(1/\sqrt{n})$ with proportionality constants that depend on the confidence $\delta$, and other constants describing the complexity of feature mapping $\Phi$ such as its dimensionality $m$, the difference between its maximum and minimum values $\mathbf{d}$, and bounds for the solutions of (6) with vectors $\mathbf{a}$ in the convex hull of $\Phi(\mathcal{X} \times \mathcal{Y})$.

The generalization bounds for the risk provided in Theorem 3 of [5] and Theorems 2 and 3 of [4] are analogous to those in inequalities (9) and (11) above. In particular, they also show risk's bounds w.r.t. to the minimax risk corresponding to an infinite number of samples. The bounds in [5] and [4] correspond to uncertainty sets defined by expectation fits with empirical marginals and Wasserstein distances, respectively, while the bounds (9) and (11) above correspond to the proposed uncertainty sets in (1). The generalization bounds in Corollary 3.2 in [9] and Theorem 2 of [11] are analogous to those in inequalities (9) and (10) above. In particular, they also show how the risk can be upper bounded (assymptotically in [9] and inequality (10) or with certain confidence in [11] and inequality (9)) by the corresponding finite-sample minimax expected loss. The bounds in [9] and [11] correspond with uncertainty sets defined by f-divergences, and Wasserstein distances, respectively, while the bounds (9) and (10) above correspond with the proposed uncertainty sets defined by linear constraints.

## 4 Implementation of MRCs

Algorithm 1 describes MRCs learning stage that obtains parameters $\boldsymbol{\mu}^*, \nu^*$ by solving optimization problem (3) in Theorem 1 given expectation estimates in (2) obtained from training samples. An upper bound for the expected loss is directly obtained as by-product of such optimization while a lower bound for the expected loss requires to solve an additional linear optimization problem given by (7) in Theorem 2.

**Algorithm 1** – Pseudocode for MRC learning

---

**Input:** Training samples $(x_1, y_1), (x_2, y_2), \ldots, (x_n, y_n)$, width of confidence intervals $\boldsymbol{\lambda}$
feature mapping $\Phi$, and matrices $\boldsymbol{\Phi}_1, \boldsymbol{\Phi}_2, \ldots, \boldsymbol{\Phi}_r$ satisfying (12)

**Output:** Parameters $\boldsymbol{\mu}^*, \nu^*$, upper bound $R^{\mathbf{a}_n, \mathbf{b}_n}$, and [Optional] lower bound $L^{\mathbf{a}_n, \mathbf{b}_n}$

1: $\boldsymbol{\tau}_n \leftarrow \frac{1}{n}\sum_{i=1}^n \Phi(x_i, y_i)$, $\mathbf{a}_n \leftarrow \boldsymbol{\tau}_n - \boldsymbol{\lambda}\frac{1}{\sqrt{n}}$, $\mathbf{b}_n \leftarrow \boldsymbol{\tau}_n + \boldsymbol{\lambda}\frac{1}{\sqrt{n}}$

2: $\boldsymbol{\mu}_b^*, \boldsymbol{\mu}_a^*, \nu^* \leftarrow \underset{\boldsymbol{\mu}_a, \boldsymbol{\mu}_b, \nu}{\arg\min} \quad \mathbf{b}_n^{\mathrm{T}}\boldsymbol{\mu}_b - \mathbf{a}_n^{\mathrm{T}}\boldsymbol{\mu}_a - \nu$

$\qquad\qquad$ s. t. $\;(\mathbf{1}_{\mathcal{C}})^{\mathrm{T}}\left(\boldsymbol{\Phi}_i(\boldsymbol{\mu}_a - \boldsymbol{\mu}_b) + \nu\mathbf{1}\right) \leq 1 - |\mathcal{C}|, \; \forall i \in \{1, 2, \ldots, r\}, \mathcal{C} \subseteq \mathcal{Y}, \mathcal{C} \neq \emptyset$

$\qquad\qquad\qquad \boldsymbol{\mu}_a, \boldsymbol{\mu}_b \succeq \mathbf{0}$

3: $\boldsymbol{\mu}^* \leftarrow \boldsymbol{\mu}_a^* - \boldsymbol{\mu}_b^*$, $R^{\mathbf{a}_n, \mathbf{b}_n} \leftarrow \mathbf{b}_n^{\mathrm{T}}\boldsymbol{\mu}_b^* - \mathbf{a}_n^{\mathrm{T}}\boldsymbol{\mu}_a^* - \nu^*$

4: [Optional] $L^{\mathbf{a}_n, \mathbf{b}_n} \leftarrow - \underset{\boldsymbol{\mu}_a, \boldsymbol{\mu}_b, \nu}{\min} \quad \mathbf{b}_n^{\mathrm{T}}\boldsymbol{\mu}_b - \mathbf{a}_n^{\mathrm{T}}\boldsymbol{\mu}_a - \nu$

$\qquad\qquad$ s. t. $\quad \boldsymbol{\Phi}_i(\boldsymbol{\mu}_a - \boldsymbol{\mu}_b) + \nu\mathbf{1} \preceq \mathbf{1} - \boldsymbol{\varepsilon}_i, \; \forall i \in \{1, 2, \ldots, r\}$

$\qquad\qquad\qquad \boldsymbol{\mu}_a, \boldsymbol{\mu}_b \succeq \mathbf{0}$

where $\boldsymbol{\varepsilon}_i = \begin{cases} (\boldsymbol{\Phi}_i\boldsymbol{\mu}^* + (\nu^* + 1)\mathbf{1}])_+/c_i & \text{if } c_i \neq 0 \\ \mathbf{1}/|\mathcal{Y}| & \text{if } c_i = 0 \end{cases}$ and $c_i = \|(\boldsymbol{\Phi}_i\boldsymbol{\mu}^* + (\nu^* + 1)\mathbf{1})_+\|_1$

---

Optimization problems (3) and (7) addressed at learning can be efficiently solved; in the following we show equivalent representations of such optimization problems that are appropriate for implementation. For each $x \in \mathcal{X}$, let $\boldsymbol{\Phi}_x$ be the $|\mathcal{Y}| \times m$ matrix with $y$-th row equal to $\Phi(x, y)^{\mathrm{T}}$. If $\boldsymbol{\Phi}_1, \boldsymbol{\Phi}_2, \ldots, \boldsymbol{\Phi}_r$ are $r$ matrices describing the range of matrices $\boldsymbol{\Phi}_x$ for varying $x \in \mathcal{X}$, i.e.,

$$\{\boldsymbol{\Phi}_i : \; i = 1, 2, \ldots, r\} = \{\boldsymbol{\Phi}_x : \; x \in \mathcal{X}\} \tag{12}$$

then, constraints in optimization problem (7) are equivalent to $2m + r|\mathcal{Y}|$ linear constraints. Constraints in optimization problem (3) are equivalent to $2m$ linear and $r$ nonlinear constraints since $\|(\boldsymbol{\Phi}(\boldsymbol{\mu}_a - \boldsymbol{\mu}_b) + (\nu + 1)\mathbf{1})_+\|_{1,\infty} \leq 1$ is equivalent to

$$\|\left(\boldsymbol{\Phi}_i(\boldsymbol{\mu}_a - \boldsymbol{\mu}_b) + (\nu + 1)\mathbf{1}\right)_+\|_1 \leq 1 \text{ for } i = 1, 2, \ldots, r. \tag{13}$$

Furthermore, constraints in optimization problem (3) are also equivalent to $2m + r(2^{|\mathcal{Y}|} - 1)$ linear constraints because (13) is equivalent to

$$(\mathbf{1}_{\mathcal{C}})^{\mathrm{T}}\left(\boldsymbol{\Phi}_i(\boldsymbol{\mu}_a - \boldsymbol{\mu}_b) + \nu\mathbf{1}\right) \leq 1 - |\mathcal{C}|, \; \forall \, i \in \{1, 2, \ldots, r\}, \; \mathcal{C} \subseteq \mathcal{Y}, \mathcal{C} \neq \emptyset$$

since $\|(\boldsymbol{\Phi}_i(\boldsymbol{\mu}_a - \boldsymbol{\mu}_b) + (\nu + 1)\mathbf{1})_+\|_1 = \underset{\mathcal{C} \subseteq \mathcal{Y}}{\max}(\mathbf{1}_{\mathcal{C}})^{\mathrm{T}}\left(\boldsymbol{\Phi}_i(\boldsymbol{\mu}_a - \boldsymbol{\mu}_b) + (\nu + 1)\mathbf{1}\right).$

Classification problems with a moderate number of classes $|\mathcal{Y}|$ can benefit by the formulation of (3) as a linear optimization problem with $2m + r(2^{|\mathcal{Y}|} - 1)$ constraints instead of that as nonlinear convex optimization with $2m + r$ constraints. The number $r$ of matrices $\boldsymbol{\Phi}_1, \boldsymbol{\Phi}_2, \ldots, \boldsymbol{\Phi}_r$ needed to cover the range of matrices $\boldsymbol{\Phi}_x$, $x \in \mathcal{X}$, determines the number of constraints in the optimization problems solved for MRC learning. Efficient optimization can be achieved using constraint generation techniques or approximations with a subset of constraints.

At prediction stage, MRCs use the parameters $\boldsymbol{\mu}^*$ and $\nu^*$ obtained at learning to assign label $y \in \mathcal{Y}$ to instance $x \in \mathcal{X}$ with probability

$$\mathrm{h}^{\mathbf{a}, \mathbf{b}}(y|x) = \begin{cases} (\Phi(x, y)^{\mathrm{T}}\boldsymbol{\mu}^* + \nu^* + 1)_+/c_x & \text{if } c_x \neq 0 \\ 1/\mathcal{Y} & \text{if } c_x = 0 \end{cases} \tag{14}$$

that satisfies (4) in Theorem 1 by taking $c_x = \sum_{y \in \mathcal{Y}}(\Phi(x, y)^{\mathrm{T}}\boldsymbol{\mu}^* + \nu^* + 1)_+.$

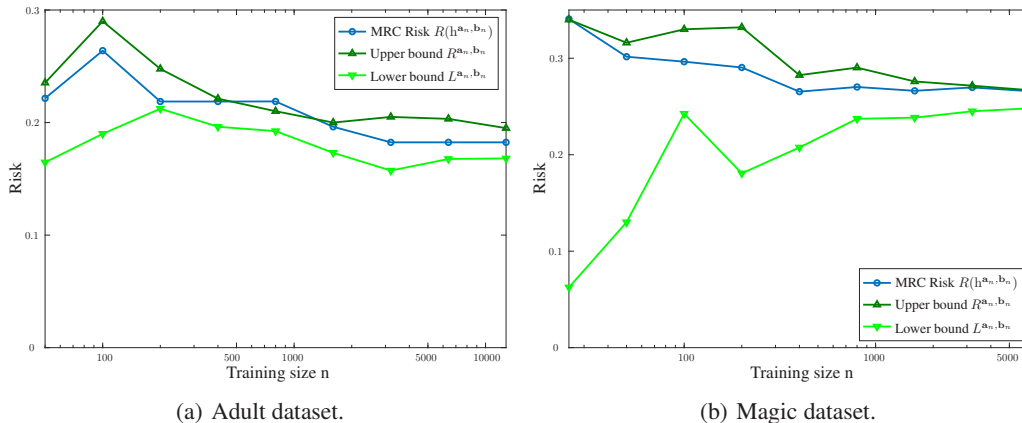

| (a) Adult dataset. | (b) Magic dataset. |

Figure 1: Upper and lower MRC risk bounds obtained at learning.

## 5   Experimental results

In this section we show numerical results for MRCs using 8 UCI datasets for multi-class classification. The first set of results shows the suitability of the upper and lower bounds $R^{\mathbf{a},\mathbf{b}}$ and $L^{\mathbf{a},\mathbf{b}}$ for MRCs with varying training sizes, while the second set of results compares the classification error of MRCs w.r.t. state-of-the-art techniques.

MRCs' results are obtained using feature mappings given by instances' thresholding, similarly to those used by maximum entropy and logistic regression methods [13,16,17]. Such feature mappings are adequate for a streamlined implementation of MRCs because they take a reduced number of values.[1] Let each instance $x \in \mathcal{X}$ be given by $\mathbf{x} = [x^{(1)}, x^{(2)}, \ldots, x^{(D)}]^{\mathrm{T}} \in \mathbb{R}^D$, and let $\mathrm{Th}_i \in \mathbb{R}$ be a threshold corresponding with dimension $d_i \in \{1, 2, \ldots, D\}$ for $i = 1, 2, \ldots, k$. We consider feature mappings with $m = |\mathcal{Y}|(k+1)$ components corresponding to the different combinations of labels and thresholds. Specifically,

$$\Phi^{(l)}(x,y) = \mathbb{I}\{y = i\} \text{ for } l = (i-1)(k+1) + 1, \ i = 1, 2, \ldots, |\mathcal{Y}|$$
$$\Phi^{(l)}(x,y) = \mathbb{I}\big\{x^{(d_j)} \leq \mathrm{Th}_j\big\}\mathbb{I}\{y = i\}$$
$$\text{for } l = (i-1)(k+1) + j + 1, \ i = 1, 2, \ldots, |\mathcal{Y}|, j = 1, 2, \ldots, k. \quad (15)$$

We obtain up to $k = 200/|\mathcal{Y}|$ thresholds using one-dimensional decision trees (decision stumps) so that the feature mapping has up to $m = 200 + |\mathcal{Y}|$ components, and we solve the optimization problems at learning with the constraints corresponding to the $r = n$ matrices $\mathbf{\Phi}_i = \mathbf{\Phi}_{x_i}, i = 1, 2, \ldots, n$, obtained from the $n$ training instances. For all datasets, interval estimates for feature mapping expectations were obtained using (2) with $\lambda^{(i)} = 0.25$ for $i = 1, 2, \ldots, m$. All other classification techniques were implemented using their default parameters, and the convex optimization problems have been solved using CVX package [18].

In the first set of experimental results, we use "Adult" and "Magic" data sets from the UCI repository. For each training size, one instantiation of training samples is used for learning as described in Algorithm 1, and MRC's risk is estimated using the remaining samples. It can be observed from the Figures 1(a) and 1(b) that the lower and upper bounds obtained at learning can offer accurate estimates for the risk without using test samples.

In the second set of experimental results, we use 6 data sets from the UCI repository (first column of Table 1). MRCs are compared with 7 classifiers: decision tree (DT), quadratic discriminant analysis (QDA), k-nearest neighbor (KNN), Gaussian kernel SVM, and random forest (RF), as well as the related RRM classifiers adversarial multiclass classifier (AMC), and maximum entropy machine (MEM). The first 5 classifiers were implemented using scikit-learn package, AMC [7] was implemented with Gaussian kernel using the publicly available code

Table 1: Classification error and performance bounds of MRC in comparison with state-of-the-art techniques.

| Data set | LB | MRC | UB | QDA | DT | KNN | SVM | RF | AMC | MEM |
|---|---|---|---|---|---|---|---|---|---|---|
| Mammog. | .16 | $.18 \pm .04$ | .21 | $.20 \pm .04$ | $.24 \pm .04$ | $.22 \pm .04$ | $.18 \pm .03$ | $.21 \pm .06$ | $.18 \pm .03$ | $.22 \pm .04$ |
| Haberman | .24 | $.27 \pm .03$ | .27 | $.24 \pm .03$ | $.39 \pm .14$ | $.30 \pm .07$ | $.26 \pm .04$ | $.35 \pm .12$ | $.25 \pm .04$ | $.27 \pm .02$ |
| Indian liv. | .28 | $.29 \pm .01$ | .30 | $.44 \pm .08$ | $.35 \pm .09$ | $.34 \pm .05$ | $.29 \pm .02$ | $.30 \pm .05$ | $.29 \pm .01$ | $.29 \pm .01$ |
| Diabetes | .22 | $.26 \pm .03$ | .28 | $.26 \pm .03$ | $.29 \pm .07$ | $.26 \pm .05$ | $.24 \pm .04$ | $.26 \pm .05$ | $.24 \pm .04$ | $.34 \pm .04$ |
| Credit | .12 | $.15 \pm .18$ | .17 | $.22 \pm .07$ | $.22 \pm .14$ | $.14 \pm .09$ | $.16 \pm .17$ | $.17 \pm .15$ | $.15 \pm .18$ | $.14 \pm .04$ |
| Glass | .22 | $.36 \pm .08$ | .47 | $.64 \pm .04$ | $.39 \pm .18$ | $.34 \pm .08$ | $.34 \pm .11$ | $.40 \pm .14$ | $.42 \pm .14$ | $.35 \pm .08$ |
| Avg. rank | | 2.7 | | 5.1 | 7.0 | 3.8 | 2.0 | 5.3 | 2.5 | 3.8 |

provided by the authors in `https://github.com/rizalzaf/adversarial-multiclass`, and MEM was implemented as shown in [5]. The errors and standard deviations in Table 1 have been estimated using paired and stratified 10-fold cross validation. The upper and lower bounds showed in columns UB and LB, respectively, are obtained without averaging, that is, by one-time learning MRCs with all samples. It can be observed from the table that the accuracy of proposed MRCs is competitive with state-of-the-art techniques even using a simple feature mapping given by instances' thresholding. Table 1 also shows the tightness of the presented performance bounds for assorted datasets. Python code with the proposed MRC is provided in `https://github.com/MachineLearningBCAM/Minimax-risk-classifiers-NeurIPS-2020` with the settings used in these experimental results.

## 6 Conclusion

The proposed MRCs minimize the worst-case expected 0-1 loss over general classification rules, and provide performance guarantees at learning. The paper also describes MRCs' implementation in practice, and presents their finite-sample generalization bounds. Experimentation with benchmark datasets shows the reliability and tightness of the presented performance bounds, and the competitive classification performance of MRCs with simple feature mappings given by thresholds. The results presented show that supervised classification does not require to choose a surrogate loss that substitutes original 0-1 loss, and a specific family that constraints classification rules. Differently from conventional techniques, the inductive bias exploited by MRCs comes only from a feature mapping that serves to constrain the distributions considered. Learning with MRCs is achieved without further design choices by solving linear optimization problems that can also provide tight performance guarantees.

## Broader Impact

The results presented in the paper can enable new approaches for supervised learning that can benefit general applications of supervised classification. Such results do not put anybody at a disadvantage, create consequences in case of failure or leverage biases in the data.

## Acknowledgments and Disclosure of Funding

Funding in direct support of this work has been provided by the Spanish Ministry of Economy and Competitiveness MINECO through Ramon y Cajal Grant RYC-2016-19383, BCAM's Severo Ochoa Excellence Accreditation SEV-2017-0718, Project PID2019-105058GA-I00, and Project TIN2017-82626-R, and by the Basque Government through the ELKARTEK and BERC 2018-2021 programmes.

## Footnotes

[1]The implementation of MRCs with more sophisticated feature mappings, such as those embedding data into a RKHS, can be enabled by using constraint generation techniques or subgradient descent methods.

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
