[Supplementary Material]

# Appendices

## A   Auxiliary lemmas

The proofs of Theorem 1 and Theorem 2 require the lemmas provided below.

**Lemma 1.** The norms $\|\cdot\|_{\infty,1}$ and $\|\cdot\|_{1,\infty}$ are dual.

*Proof.* The dual norm of $\|\cdot\|_{\infty,1}$ assigns each $\mathbf{w} \in \mathbb{R}^{|\mathcal{I}||\mathcal{J}|}$ for finite sets $\mathcal{I}$ and $\mathcal{J}$, the real number

$$\sup_{\mathbf{v}: \|\mathbf{v}\|_{\infty,1}\leq 1} \mathbf{w}^\mathsf{T}\mathbf{v}.$$

We have that for $\mathbf{v}$ with $\|\mathbf{v}\|_{\infty,1} \leq 1$

$$\mathbf{w}^\mathsf{T}\mathbf{v} = \sum_{i\in\mathcal{I}}\sum_{j\in\mathcal{J}} w_{(i,j)}v_{(i,j)} \leq \sum_{i\in\mathcal{I}}\sum_{j\in\mathcal{J}} |w_{(i,j)}||v_{(i,j)}|$$

$$\leq \sum_{i\in\mathcal{I}}\left(\max_j |v_{(i,j)}|\right)\sum_{j\in\mathcal{J}} |w_{(i,j)}| \leq \max_{i\in\mathcal{I}}\sum_{j\in\mathcal{J}} |w_{(i,j)}|\sum_{i\in\mathcal{I}}\left(\max_j |v_{(i,j)}|\right)$$

$$= \|\mathbf{w}\|_{1,\infty}\|\mathbf{v}\|_{\infty,1} \leq \|\mathbf{w}\|_{1,\infty}$$

So, to prove the result we just need to find a vector $\mathbf{u}$ such that $\|\mathbf{u}\|_{\infty,1} \leq 1$ and $\mathbf{w}^\mathsf{T}\mathbf{u} = \|\mathbf{w}\|_{1,\infty}$. Let $\iota \in \arg\max_{i\in\mathcal{I}}\sum_{j\in\mathcal{J}} |w_{(i,j)}|$, then $\mathbf{u}$ given by

$$u_{(i,j)} = \begin{cases} 1 & \text{if } i = \iota \text{ and } w_{(i,j)} \geq 0 \\ -1 & \text{if } i = \iota \text{ and } w_{(i,j)} < 0 \\ 0 & \text{otherwise} \end{cases}$$

satisfies $\|\mathbf{u}\|_{\infty,1} \leq 1$ and $\mathbf{w}^\mathsf{T}\mathbf{u} = \|\mathbf{w}\|_{1,\infty}$.

$\square$

**Lemma 2.** Let $\mathbf{u} \in \mathbb{R}^{|\mathcal{I}||\mathcal{J}|}$ for finite sets $\mathcal{I}$ and $\mathcal{J}$, and $f_1$, $f_2$ be the functions $f_1(\mathbf{v}) = \|\mathbf{v}\|_{\infty,1} - \mathbf{1}^\mathsf{T}\mathbf{v} + I_+(\mathbf{v})$ and $f_2(\mathbf{v}) = \mathbf{v}^\mathsf{T}\mathbf{u} + I_+(\mathbf{v})$ for $\mathbf{v} \in \mathbb{R}^{|\mathcal{I}||\mathcal{J}|}$, where

$$I_+(\mathbf{v}) = \begin{cases} 0 & \text{if } \mathbf{v} \succeq \mathbf{0} \\ \infty & \text{otherwise} \end{cases}.$$

Then, their conjugate functions are

$$f_1^*(\mathbf{w}) = \begin{cases} 0 & \text{if } \|(\mathbf{1} + \mathbf{w})_+\|_{1,\infty} \leq 1 \\ \infty & \text{otherwise} \end{cases}$$

$$f_2^*(\mathbf{w}) = \begin{cases} 0 & \text{if } \mathbf{w} \preceq \mathbf{u} \\ \infty & \text{otherwise} \end{cases}.$$

*Proof.* By definition of conjugate function we have

$$f_1^*(\mathbf{w}) = \sup_{\mathbf{v}}(\mathbf{w}^\mathsf{T}\mathbf{v} - \|\mathbf{v}\|_{\infty,1} + \mathbf{1}^\mathsf{T}\mathbf{v} - I_+(\mathbf{v})) = \sup_{\mathbf{v}\succeq 0}((\mathbf{1} + \mathbf{w})^\mathsf{T}\mathbf{v} - \|\mathbf{v}\|_{\infty,1}).$$

- If $\|(\mathbf{1} + \mathbf{w})_+\|_{1,\infty} \leq 1$, for each $\mathbf{v} \succeq \mathbf{0}$, $\mathbf{v} \neq \mathbf{0}$ we have

$$(\mathbf{1} + \mathbf{w})^\mathsf{T}\mathbf{v} \leq ((\mathbf{1} + \mathbf{w})_+)^\mathsf{T}\mathbf{v} = \|\mathbf{v}\|_{\infty,1}\left(((\mathbf{1} + \mathbf{w})_+)^\mathsf{T}\frac{\mathbf{v}}{\|\mathbf{v}\|_{\infty,1}}\right)$$

and by definition of dual norm we get

$$(\mathbf{1} + \mathbf{w})^\mathsf{T}\mathbf{v} \leq \|\mathbf{v}\|_{\infty,1}\|(\mathbf{1} + \mathbf{w})_+\|_{1,\infty} \leq \|\mathbf{v}\|_{\infty,1}$$

which implies

$$(\mathbf{1} + \mathbf{w})^\mathsf{T}\mathbf{v} - \|\mathbf{v}\|_{\infty,1} \leq 0.$$

Moreover, $(\mathbf{1} + \mathbf{w})^\mathsf{T}\mathbf{0} - \|\mathbf{0}\|_{\infty,1} = 0$, so we have that $f_1^*(\mathbf{w}) = 0$.

- If $\|(\mathbf{1} + \mathbf{w})_+\|_{1,\infty} > 1$, by definition of dual norm and using Lemma 1 there exists $\mathbf{u}$ such that $((\mathbf{1} + \mathbf{w})_+)^{\mathsf{T}}\mathbf{u} > 1$ and $\|\mathbf{u}\|_{\infty,1} \leq 1$. Define $\tilde{\mathbf{u}}$ as

$$\tilde{u}_{(i,j)} = \begin{cases} u_{(i,j)} & \text{if } u_{(i,j)} \geq 0 \text{ and } 1 + w_{(i,j)} \geq 0 \\ 0 & \text{if } u_{(i,j)} < 0 \text{ or } 1 + w_{(i,j)} < 0 \end{cases}$$

By definition of $\tilde{\mathbf{u}}$ and $\|\cdot\|_{\infty,1}$ we have

$$\|\tilde{\mathbf{u}}\|_{\infty,1} \leq \|\mathbf{u}\|_{\infty,1} \leq 1$$

and

$$(\mathbf{1} + \mathbf{w})^{\mathsf{T}}\tilde{\mathbf{u}} = ((\mathbf{1} + \mathbf{w})_+)^{\mathsf{T}}\tilde{\mathbf{u}} \geq ((\mathbf{1} + \mathbf{w})_+)^{\mathsf{T}}\mathbf{u} > 1.$$

Now let $t > 0$ and take $\mathbf{v} = t\tilde{\mathbf{u}} \succeq \mathbf{0}$, then we have

$$(\mathbf{1} + \mathbf{w})^{\mathsf{T}}\mathbf{v} - \|\mathbf{v}\|_{\infty,1} = t\left((\mathbf{1} + \mathbf{w})^{\mathsf{T}}\tilde{\mathbf{u}} - \|\tilde{\mathbf{u}}\|_{\infty,1}\right)$$

which tends to infinity as $t \to +\infty$ because $(\mathbf{1} + \mathbf{w})^{\mathsf{T}}\tilde{\mathbf{u}} - \|\tilde{\mathbf{u}}\|_{\infty,1} > 0$, so we have that $f_1^*(\mathbf{w}) = +\infty$.

Finally, the expression for $f_2^*$ is straightforward since

$$f_2^*(\mathbf{w}) = \sup_{\mathbf{v} \succeq \mathbf{0}}((\mathbf{w} - \mathbf{u})^{\mathsf{T}}\mathbf{v}).$$

$\square$

# B   Proof of Theorem 1

Let set $\widetilde{\mathcal{U}}$ and function $\widetilde{\ell}(\mathrm{h}, \mathrm{p})$ be given by

$$\widetilde{\mathcal{U}} = \{\mathrm{p} : \mathcal{X} \times \mathcal{Y} \to \mathbb{R} \text{ s.t. } \mathrm{p} \succeq \mathbf{0}, \ \|\mathrm{p}\|_{1,\infty} \leq 1\}$$

$$\widetilde{\ell}(\mathrm{h}, \mathrm{p}) = \mathbf{b}^{\mathsf{T}}\boldsymbol{\mu}_b^* - \mathbf{a}^{\mathsf{T}}\boldsymbol{\mu}_a^* - \nu^* + \mathrm{p}^{\mathsf{T}}(\boldsymbol{\Phi}(\boldsymbol{\mu}_a^* - \boldsymbol{\mu}_b^*) + (\nu^* + 1)\mathbf{1} - \mathrm{h}).$$

In the first step of the proof we show that $\mathrm{h}^{\mathbf{a},\mathbf{b}}$ satisfying (4) is a solution of optimization problem $\min_{\mathrm{h} \in T(\mathcal{X},\mathcal{Y})} \max_{\mathrm{p} \in \widetilde{\mathcal{U}}} \widetilde{\ell}(\mathrm{h}, \mathrm{p})$, and in the second step of the proof we show that a solution of $\min_{\mathrm{h} \in T(\mathcal{X},\mathcal{Y})} \max_{\mathrm{p} \in \widetilde{\mathcal{U}}} \widetilde{\ell}(\mathrm{h}, \mathrm{p})$ is also a solution of $\min_{\mathrm{h} \in T(\mathcal{X},\mathcal{Y})} \max_{\mathrm{p} \in \mathcal{U}^{\mathbf{a},\mathbf{b}}} \ell(\mathrm{h}, \mathrm{p})$.

For the first step, note that

$$\widetilde{\ell}(\mathrm{h}, \mathrm{p}) = \mathbf{b}^{\mathsf{T}}\boldsymbol{\mu}_b^* - \mathbf{a}^{\mathsf{T}}\boldsymbol{\mu}_a^* - \nu^* + \sum_{x \in \mathcal{X}} \mathrm{p}_x^{\mathsf{T}}\left(\boldsymbol{\Phi}_x(\boldsymbol{\mu}_a^* - \boldsymbol{\mu}_b^*) + (\nu^* + 1)\mathbf{1} - \mathrm{h}_x\right).$$

Then, optimization problem $\min_{\mathrm{h} \in T(\mathcal{X},\mathcal{Y})} \max_{\mathrm{p} \in \widetilde{\mathcal{U}}} \widetilde{\ell}(\mathrm{h}, \mathrm{p})$ is equivalent to

$$\min_{\mathrm{h}_x \in \Delta(\mathcal{Y}) \ \forall x \in \mathcal{X}} \quad \max_{\mathrm{p}_x \succeq \mathbf{0}, \|\mathrm{p}_x\|_1 \leq 1 \forall x \in \mathcal{X}} \quad \sum_{x \in \mathcal{X}} \mathrm{p}_x^{\mathsf{T}}\left(\boldsymbol{\Phi}_x(\boldsymbol{\mu}_a^* - \boldsymbol{\mu}_b^*) + (\nu^* + 1)\mathbf{1} - \mathrm{h}_x\right)$$

that is separable and has solution given by

$$\mathrm{h}_x^{\mathbf{a},\mathbf{b}} \in \arg\min_{\mathrm{h}_x \in \Delta(\mathcal{Y})} \quad \max_{\mathrm{p}_x \succeq \mathbf{0}, \|\mathrm{p}_x\|_1 \leq 1} \quad \mathrm{p}_x^{\mathsf{T}}\left(\boldsymbol{\Phi}_x(\boldsymbol{\mu}_a^* - \boldsymbol{\mu}_b^*) + (\nu^* + 1)\mathbf{1} - \mathrm{h}_x\right)$$

for each $x \in \mathcal{X}$. The inner maximization above is given in closed-form by

$$\max_{\mathrm{p}_x \succeq \mathbf{0}, \|\mathrm{p}_x\|_1 \leq 1} \mathrm{p}_x^{\mathsf{T}}\left(\boldsymbol{\Phi}_x(\boldsymbol{\mu}_a^* - \boldsymbol{\mu}_b^*) + (\nu^* + 1)\mathbf{1} - \mathrm{h}_x\right)$$

$$= \|\left(\boldsymbol{\Phi}_x(\boldsymbol{\mu}_a^* - \boldsymbol{\mu}_b^*) + (\nu^* + 1)\mathbf{1} - \mathrm{h}_x\right)_+\|_\infty \geq 0$$

that takes its minimum value 0 for any $\mathrm{h}_x^{\mathbf{a},\mathbf{b}} \succeq \boldsymbol{\Phi}_x(\boldsymbol{\mu}_a^* - \boldsymbol{\mu}_b^*) + (\nu^* + 1)\mathbf{1}$.

For the second step, if $\mathrm{h}^{\mathbf{a},\mathbf{b}}$ is a solution of $\min_{\mathrm{h} \in T(\mathcal{X},\mathcal{Y})} \max_{\mathrm{p} \in \widetilde{\mathcal{U}}} \widetilde{\ell}(\mathrm{h}, \mathrm{p})$ we have that

$$\min_{\mathrm{h} \in T(\mathcal{X},\mathcal{Y})} \max_{\mathrm{p} \in \widetilde{\mathcal{U}}} \widetilde{\ell}(\mathrm{h}, \mathrm{p}) = \max_{\mathrm{p} \in \widetilde{\mathcal{U}}} \widetilde{\ell}(\mathrm{h}^{\mathbf{a},\mathbf{b}}, \mathrm{p}) \geq \max_{\mathrm{p} \in \mathcal{U}^{\mathbf{a},\mathbf{b}}} \ell(\mathrm{h}^{\mathbf{a},\mathbf{b}}, \mathrm{p}) \geq \min_{\mathrm{h} \in T(\mathcal{X},\mathcal{Y})} \max_{\mathrm{p} \in \mathcal{U}^{\mathbf{a},\mathbf{b}}} \ell(\mathrm{h}, \mathrm{p}) \quad (16)$$

where the first inequality is due to the fact that $\mathcal{U}^{\mathbf{a},\mathbf{b}} \subset \widetilde{\mathcal{U}}$ and $\widetilde{\ell}(h,p) \geq \ell(h,p)$ for $p \in \mathcal{U}^{\mathbf{a},\mathbf{b}}$ because

$$\mathbf{b}^{\mathsf{T}}\boldsymbol{\mu}_b^* - \mathbf{a}^{\mathsf{T}}\boldsymbol{\mu}_a^* + \mathbf{p}^{\mathsf{T}}\boldsymbol{\Phi}(\boldsymbol{\mu}_a^* - \boldsymbol{\mu}_b^*) \leq 0$$

by definition of $\mathcal{U}^{\mathbf{a},\mathbf{b}}$ and since $\boldsymbol{\mu}_a^*, \boldsymbol{\mu}_b^* \succeq \mathbf{0}$.

Since $\ell(h,p)$ is continuous and convex-concave, and both $\mathcal{U}^{\mathbf{a},\mathbf{b}}$ and $T(\mathcal{X},\mathcal{Y})$ are convex and compact, the min and the max in $R^{\mathbf{a},\mathbf{b}} = \min_{h \in T(\mathcal{X},\mathcal{Y})} \max_{p \in \mathcal{U}^{\mathbf{a},\mathbf{b}}} \ell(h,p)$ can be interchanged (see e.g., [14]) and we have that $R^{\mathbf{a},\mathbf{b}} = \max_{p \in \mathcal{U}^{\mathbf{a},\mathbf{b}}} \min_{h \in T(\mathcal{X},\mathcal{Y})} \ell(h,p)$. In addition,

$$\min_{h \in T(\mathcal{X},\mathcal{Y})} \ell(h,p) = \min_{h \in T(\mathcal{X},\mathcal{Y})} \mathbf{p}^{\mathsf{T}}(\mathbf{1} - \mathbf{h}) = \mathbf{p}^{\mathsf{T}}\mathbf{1} - \|\mathbf{p}\|_{\infty,1}$$

because the optimization problem above is separable for $x \in \mathcal{X}$ and

$$\max_{h_x \in \Delta(\mathcal{Y})} \mathbf{p}_x^{\mathsf{T}}\mathbf{h}_x = \|\mathbf{p}_x\|_\infty. \tag{17}$$

Then $R^{\mathbf{a},\mathbf{b}} = \max_{p \in \mathcal{U}^{\mathbf{a},\mathbf{b}}} \mathbf{p}^{\mathsf{T}}\mathbf{1} - \|\mathbf{p}\|_{\infty,1}$ that can be written as

$$\begin{aligned} \max_{\mathbf{p}} \quad & \mathbf{p}^{\mathsf{T}}\mathbf{1} - \|\mathbf{p}\|_{\infty,1} - I_+(\mathbf{p}) \\ \text{s. t.} \quad & -\mathbf{p}^{\mathsf{T}}\mathbf{1} = -1 \\ & \mathbf{a} \preceq \boldsymbol{\Phi}^{\mathsf{T}}\mathbf{p} \preceq \mathbf{b} \end{aligned} \tag{18}$$

where

$$I_+(\mathbf{p}) = \begin{cases} 0 & \text{if } \mathbf{p} \succeq \mathbf{0} \\ \infty & \text{otherwise} \end{cases}$$

The Lagrange dual of the optimization problem (18) is

$$\begin{aligned} \min_{\boldsymbol{\mu}_a, \boldsymbol{\mu}_b \in \mathbb{R}^m, \nu \in \mathbb{R}} \quad & \mathbf{b}^{\mathsf{T}}\boldsymbol{\mu}_b - \mathbf{a}^{\mathsf{T}}\boldsymbol{\mu}_a - \nu + f^*\left(\boldsymbol{\Phi}(\boldsymbol{\mu}_a - \boldsymbol{\mu}_b) + \nu\mathbf{1}\right) \\ \text{s.t.} \quad & \boldsymbol{\mu}_a \succeq \mathbf{0}, \boldsymbol{\mu}_b \succeq \mathbf{0} \end{aligned} \tag{19}$$

where $f^*$ is the conjugate function of $f(\mathbf{p}) = \|\mathbf{p}\|_{\infty,1} - \mathbf{p}^{\mathsf{T}}\mathbf{1} + I_+(\mathbf{p})$ (see e.g., section 5.1.6 in [15]). Then, optimization problem (19) becomes (3) using the Lemma 2 above.

Strong duality holds between optimization problems (18) and (3) since constraints in (18) are affine. Then, if $\boldsymbol{\mu}_a^*, \boldsymbol{\mu}_b^*, \nu^*$ is a solution of (3) we have that $R^{\mathbf{a},\mathbf{b}}$ is equal to the value of

$$\max_{\mathbf{p}} \mathbf{p}^{\mathsf{T}}\mathbf{1} - \|\mathbf{p}\|_{\infty,1} - I_+(\mathbf{p}) - (\mathbf{p}^{\mathsf{T}}\boldsymbol{\Phi} - \mathbf{b}^{\mathsf{T}})\boldsymbol{\mu}_b^* + (\mathbf{p}^{\mathsf{T}}\boldsymbol{\Phi} - \mathbf{a}^{\mathsf{T}})\boldsymbol{\mu}_a^* + (\mathbf{p}^{\mathsf{T}}\mathbf{1} - 1)\nu^* \tag{20}$$

that equals

$$\max_{p \in \widetilde{\mathcal{U}}} \mathbf{p}^{\mathsf{T}}\mathbf{1} - \|\mathbf{p}\|_{\infty,1} + \mathbf{b}^{\mathsf{T}}\boldsymbol{\mu}_b^* - \mathbf{a}^{\mathsf{T}}\boldsymbol{\mu}_a^* - \nu^* + \mathbf{p}^{\mathsf{T}}\left(\boldsymbol{\Phi}(\boldsymbol{\mu}_a^* - \boldsymbol{\mu}_b^*) + \nu^*\mathbf{1}\right)$$

since a solution of the primal problem (18) belongs to $\widetilde{\mathcal{U}}$ and is also a solution of (20). Therefore,

$$\begin{aligned} R^{\mathbf{a},\mathbf{b}} &= \max_{p \in \widetilde{\mathcal{U}}} \min_{h \in T(\mathcal{X},\mathcal{Y})} \ell(h,p) + \mathbf{b}^{\mathsf{T}}\boldsymbol{\mu}_b^* - \mathbf{a}^{\mathsf{T}}\boldsymbol{\mu}_a^* - \nu^* + \mathbf{p}^{\mathsf{T}}\left(\boldsymbol{\Phi}(\boldsymbol{\mu}_a^* - \boldsymbol{\mu}_b^*) + \nu^*\mathbf{1}\right) \\ &= \max_{p \in \widetilde{\mathcal{U}}} \min_{h \in T(\mathcal{X},\mathcal{Y})} \widetilde{\ell}(h,p) = \min_{h \in T(\mathcal{X},\mathcal{Y})} \max_{p \in \widetilde{\mathcal{U}}} \widetilde{\ell}(h,p) \end{aligned}$$

where the last equality is due to the fact that $\widetilde{\ell}(h,p)$ is continuous and convex-concave, and both $\widetilde{\mathcal{U}}$ and $T(\mathcal{X},\mathcal{Y})$ are convex and compact. Then, inequalities in (16) are in fact equalities and $h^{\mathbf{a},\mathbf{b}}$ is solution of $\min_{h \in T(\mathcal{X},\mathcal{Y})} \max_{p \in \mathcal{U}^{\mathbf{a},\mathbf{b}}} \ell(h,p)$.

## C  Proof of Theorem 2

The result is a direct consequence of the fact that for any $p \in \mathcal{U}^{\mathbf{a},\mathbf{b}}$

$$\min_{\widetilde{p} \in \mathcal{U}^{\mathbf{a},\mathbf{b}}} \ell(h,\widetilde{p}) \leq \ell(h,p) \leq \max_{\widetilde{p} \in \mathcal{U}^{\mathbf{a},\mathbf{b}}} \ell(h,\widetilde{p})$$

and

$$\min_{\widetilde{p} \in \mathcal{U}^{a,b}} \ell(h, \widetilde{p}) = \min_{\widetilde{p} \in \mathcal{U}^{a,b}} \widetilde{p}^T(1-h)$$

$$\max_{\widetilde{p} \in \mathcal{U}^{a,b}} \ell(h, \widetilde{p}) = -\min_{\widetilde{p} \in \mathcal{U}^{a,b}} \widetilde{p}^T(h-1).$$

The expression for $\kappa^{a,b}(q)$ in (7) is obtained since

$$
\begin{aligned}
\min_{\widetilde{p} \in \mathcal{U}^{a,b}} \widetilde{p}^T(-q) = \quad &\min_{\widetilde{p}} \quad \widetilde{p}^T(-q) + I_+(\widetilde{p}) \\
&\text{s. t.} \quad -1^T\widetilde{p} = -1 \\
&\qquad\quad a \preceq \Phi^T\widetilde{p} \preceq b
\end{aligned}
\tag{21}
$$

where

$$I_+(\widetilde{p}) = \begin{cases} 0 & \text{if } \widetilde{p} \succeq 0 \\ \infty & \text{otherwise} \end{cases}$$

Then, the Lagrange dual of the optimization problem (21) is

$$
\begin{aligned}
\max_{\mu_a, \mu_b \in \mathbb{R}^m, \nu \in \mathbb{R}} \quad & a^T\mu_a - b^T\mu_b + \nu - f^*\left(\Phi(\mu_a - \mu_b) + \nu 1\right) \\
\text{s.t.} \quad & \mu_a \succeq 0, \mu_b \succeq 0
\end{aligned}
\tag{22}
$$

where $f^*$ is the conjugate function of $f(\widetilde{p}) = \widetilde{p}^T(-q) + I^+(\widetilde{p})$ that leads to (7) using Lemma 2.

## D  Proof of Theorem 3

Firstly, with probability at least $1-\delta$ we have that $p^* \in \mathcal{U}^{a_n, b_n}$ and

$$\|\tau_\infty - \tau_n\|_2 \leq \|d\|_2 \sqrt{\frac{\log m + \log \frac{2}{\delta}}{2n}}$$

because, using Hoeffding's inequality [19] we have that for $i = 1, 2, \ldots, m$

$$\mathbb{P}\left\{|\tau_{\infty,i} - \tau_{n,i}| < t_i\right\} \geq 1 - 2\exp\left\{-\frac{2n^2 t_i^2}{n d_i^2}\right\}$$

so taking $t_i = d_i\sqrt{\frac{\log m + \log \frac{2}{\delta}}{2n}}$ we get

$$\mathbb{P}\left\{|\tau_{\infty,i} - \tau_{n,i}| < d_i\sqrt{\frac{\log m + \log \frac{2}{\delta}}{2n}}\right\} \geq 1 - 2\exp\left\{-\log m - \log\frac{2}{\delta}\right\} = 1 - \frac{\delta}{m}$$

and using the union bound we have that

$$\mathbb{P}\left\{|\tau_{\infty,i} - \tau_{n,i}| < d_i\sqrt{\frac{\log m + \log \frac{2}{\delta}}{2n}}, i = 1, 2, \ldots, m\right\}$$

$$\geq 1 - m + \sum_{i=1}^{m} \mathbb{P}\left\{|\tau_{\infty,i} - \tau_{n,i}| < d_i\sqrt{\frac{\log m + \log \frac{2}{\delta}}{2n}}\right\}$$

$$\geq 1 - \delta.$$

For the first inequality in (9), we have that $R(h^{a_n, b_n}) \leq R^{a_n, b_n}$ with probability at least $1-\delta$ since $p^* \in \mathcal{U}^{a_n, b_n}$ with probability at least $1-\delta$.

For the second inequality in (9), let $\mu^*, \nu^*$ be the solution with minimum euclidean norm of (6) for $a = \tau_\infty$; $[(\mu^*)^+, (-\mu^*)^+, \nu^*]$ is a feasible point of (3) because $\mu^* = (\mu^*)^+ - (-\mu^*)^+$ and $\mu^*, \nu^*$ is a feasible point of (6). Hence

$$R^{a_n, b_n} \leq b_n^T(-\mu^*)^+ - a_n^T(\mu^*)^+ - \nu^* = R^{\tau_\infty} + (b_n - \tau_\infty)^T(-\mu^*)^+ + (\tau_\infty - a_n)^T(\mu^*)^+$$

$$= R^{\tau_\infty} - \left( \tau_\infty - \tau_n - \mathbf{d}\sqrt{\frac{\log m + \log \frac{2}{\delta}}{2n}} \right)^{\mathrm{T}} (-\boldsymbol{\mu}^*)^+ + \left( \tau_\infty - \tau_n + \mathbf{d}\sqrt{\frac{\log m + \log \frac{2}{\delta}}{2n}} \right)^{\mathrm{T}} (\boldsymbol{\mu}^*)^+$$

$$= R^{\tau_\infty} + (\tau_n - \tau_\infty)^{\mathrm{T}} \boldsymbol{\mu}^* + \sqrt{\frac{\log m + \log \frac{2}{\delta}}{2n}} \mathbf{d}^{\mathrm{T}} ((\boldsymbol{\mu}^*)^+ + (-\boldsymbol{\mu}^*)^+)$$

Then the result is obtained using Cauchy-Schwarz inequality and the fact that $\|(\boldsymbol{\mu}^*)^+ + (-\boldsymbol{\mu}^*)^+\|_2 = \|\boldsymbol{\mu}^*\|_2$.

For the result in (10), note that using Theorem 2 and since $\mathrm{p}^* \in \mathcal{U}^{\mathbf{a}_n, \mathbf{b}_n}$ with probability at least $1 - \delta$ we have that

$$R(\mathrm{h}^{\tau_n}) \leq \max_{\mathrm{p} \in \mathcal{U}^{\mathbf{a}_n, \mathbf{b}_n}} \ell(\mathrm{h}^{\tau_n}, \mathrm{p}) = \min_{\boldsymbol{\Phi}(\boldsymbol{\mu}_a - \boldsymbol{\mu}_a) + \nu \mathbf{1} \preceq \mathbf{h}^{\tau_n} - \mathbf{1}} \mathbf{b}_n^{\mathrm{T}} \boldsymbol{\mu}_b - \mathbf{a}_n^{\mathrm{T}} \boldsymbol{\mu}_a - \nu$$

so that, if $\boldsymbol{\mu}_n^*, \nu_n^*$ is the solution with minimum euclidean norm of (6) for $\mathbf{a} = \tau_n$, we have that $R(\mathrm{h}^{\tau_n}) \leq \mathbf{b}_n^{\mathrm{T}} (-\boldsymbol{\mu}_n^*)^+ - \mathbf{a}_n^{\mathrm{T}} (\boldsymbol{\mu}_n^*)^+ - \nu_n^*$ because $\boldsymbol{\mu}_n^* = (\boldsymbol{\mu}_n^*)^+ - (-\boldsymbol{\mu}_n^*)^+$ and $\boldsymbol{\Phi}\boldsymbol{\mu}_n^* + \nu_n^* \mathbf{1} \preceq \mathbf{h}^{\tau_n} - \mathbf{1}$ by definition of $\mathbf{h}^{\tau_n}$. Therefore, the result is obtained since

$$R(\mathrm{h}^{\tau_n}) \leq \left( \tau_n + \mathbf{d}\sqrt{\frac{\log m + \log \frac{2}{\delta}}{2n}} \right)^{\mathrm{T}} (-\boldsymbol{\mu}_n^*)^+ - \left( \tau_n - \mathbf{d}\sqrt{\frac{\log m + \log \frac{2}{\delta}}{2n}} \right)^{\mathrm{T}} (\boldsymbol{\mu}_n^*)^+ - \nu_n^*$$

$$= R^{\tau_n} + \mathbf{d}^{\mathrm{T}} \sqrt{\frac{\log m + \log \frac{2}{\delta}}{2n}} \left( (\boldsymbol{\mu}_n^*)^+ + (-\boldsymbol{\mu}_n^*)^+ \right).$$

For the result in (11), note that using Theorem 2 and since $\mathrm{p}^* \in \mathcal{U}^{\tau_\infty}$ we have that

$$R(\mathrm{h}^{\tau_n}) \leq \max_{\mathrm{p} \in \mathcal{U}^{\tau_\infty}} \ell(\mathrm{h}^{\tau_n}, \mathrm{p}) = \min_{\boldsymbol{\Phi}\boldsymbol{\mu} + \nu \mathbf{1} \preceq \mathbf{h}^{\tau_n} - \mathbf{1}} - (\tau_\infty)^{\mathrm{T}} \boldsymbol{\mu} - \nu$$

so that, if $\boldsymbol{\mu}_n^*, \nu_n^*$ is the solution with minimum euclidean norm of (6) for $\mathbf{a} = \tau_n$, we have that $R(\mathrm{h}^{\tau_n}) \leq -(\tau_\infty)^{\mathrm{T}} \boldsymbol{\mu}_n^* - \nu_n^*$ because $\boldsymbol{\Phi}\boldsymbol{\mu}_n^* + \nu_n^* \mathbf{1} \preceq \mathbf{h}^{\tau_n} - \mathbf{1}$ by definition of $\mathbf{h}^{\tau_n}$. Let $\boldsymbol{\mu}^*, \nu^*$ be the solution with minimum euclidean norm of (6) for $\mathbf{a} = \tau_\infty$, the result is obtained since

$$R(\mathrm{h}^{\tau_n}) \leq -(\tau_\infty)^{\mathrm{T}} \boldsymbol{\mu}_n^* - \nu_n^* + \tau_n^{\mathrm{T}} \boldsymbol{\mu}_n^* - \tau_n^{\mathrm{T}} \boldsymbol{\mu}_n^* + (\tau_\infty)^{\mathrm{T}} \boldsymbol{\mu}^* + \nu^* - (\tau_\infty)^{\mathrm{T}} \boldsymbol{\mu}^* - \nu^*$$

$$= (\tau_n - \tau_\infty)^{\mathrm{T}} \boldsymbol{\mu}_n^* + R^{\tau_\infty} - \tau_n^{\mathrm{T}} \boldsymbol{\mu}_n^* - \nu_n^* + (\tau_\infty)^{\mathrm{T}} \boldsymbol{\mu}^* + \nu^*$$

$$\leq (\tau_n - \tau_\infty)^{\mathrm{T}} \boldsymbol{\mu}_n^* + (\tau_\infty - \tau_n)^{\mathrm{T}} \boldsymbol{\mu}^* + R^{\tau_\infty} \tag{23}$$

$$\leq \|\tau_n - \tau_\infty\|_2 \|\boldsymbol{\mu}_n^* - \boldsymbol{\mu}^*\|_2 + R^{\tau_\infty}$$

where (23) is due to the fact that $-\tau_n^{\mathrm{T}} \boldsymbol{\mu}_n^* - \nu_n^* \leq -\tau_n^{\mathrm{T}} \boldsymbol{\mu}^* - \nu^*$ since $\boldsymbol{\mu}^*, \nu^*$ is a feasible point of (6) for $\mathbf{a} = \tau_n$.