[Reviews · NeurIPS 2020]

Review 1

Summary and Contributions: The paper proposes a Robust Risk Minimization based classifier approach, that is optimizing 0-1 loss (instead of surrogate losses) and considers uncertainty sets that does have the true distribution.

Strengths: The approach is theoretically motivated giving performance bounds and generalization bounds. Performance bounds have been shown to hold well in practice on two datasets. Empirically, the classifier motivated as such is shown to be competitive with the other classifiers used in practice. The contribution is novel in terms of the problem setting and its analysis.

Weaknesses: It is unclear from the paper as to why optimizing for the direct 0-1 loss is important as opposed to optimizing some surrogate loss. While one can say that directly optimizing 0-1 loss should in effect be better, but the empirical results does not seem to indicate this.

Correctness: The claims and the method look correct and empirical methodology seems correct as well.

Clarity: The paper is reasonably well written. However, the theorems are not motivated using any intuition. Particularly, in theorem 1, the conditions to be satisfied by mu_a* etc. seem to have been taken out of thin air, which makes it really difficult to grasp the intuition behind the result and get a feel for why the method would work.

Relation to Prior Work: The paper discusses the prior works in the space of RRMs and explains how it differentiates itself from prior works.

Reproducibility: Yes

Additional Feedback: There were problems in installing cvxpy package to get the code running. It would be good to include some instructions in installing some of these packages or at least links where one can follow instructions to install these packages. Post Author Response: First of all, I thank the author for the detailed response, particularly to the questions raised by Reviewer 2, as that gave me lot more appreciation of the work than I had before. I am glad that you plan to include some intuition for the proofs, as it goes a long way for people like me to understand what is going on. I will retain the evaluation as before.


Review 2

Summary and Contributions: The authors present an LP-based algorithm for classification that minimizes the worst-case expected 0-1 loss across a set of distributions, called "uncertainty set." The algorithm is constructed such that the uncertainty set includes the true data distribution with a high probability. By construction, their method provides an (optimized) upper bound and they also show that lower bounds can be constructed by solving a similar LP problem. Finally, they present experimental results that support the method. ============ Post Rebuttal: I have read the authors' feedback and I am generally satisfied. I had three primary concerns and two of them were addressed in the rebuttal. The only downside I still see is that the authors do not explore the subject of choosing the mapping Phi in their paper, which I believe to be important. However, I will keep my score.

Strengths: I find the paper to be novel. The approach the authors use to formulate classification problems as a minimization of an upper bound on the true risk using uncertainty sets is quite different from the conventional methods of minimizing the empirical risk using surrogate loss functions, such as in SVM or logistic regression. In addition, their method is non-parametric and elegant. The paper itself is very well-written and a pleasure to read. Also, the experimental results are promising, particularly Figure 1, which shows that the true expectation is indeed sandwiched between the upper and lower bounds claimed in the paper.

Weaknesses: Perhaps, the biggest weakness is computational efficiency. Current classification methods are formulated as a minimization of empirical loss so they are often solved in practice using variants of the stochastic gradient descent method, which is quite efficient and scales well to massive amounts of data and labels. The proposed minimax risk classifiers (MRC) requires solving a full LP, whose time complexity is O(m^3). Regarding the evaluation in Table 1, it is not clear if the authors evaluated competing methods fairly. There is no discussion of hyper-paramter tuning. For example, if you tune the value of k using a validation dataset, kNN on Haberman achieves 22% error rate (not 30%). At the same time, however, I acknowledge that the choice of the feature map in MRC can also be optimized as well, which the authors did not since they currently use a simple thresholding rule in their evaluation.

Correctness: Aside from the concern above regarding Table 1, the claims seem to be sound. I have not checked the proofs in the appendix but the claims appear to be reasonable (for example, you would expect the log(m) to appear in Theorem 3 by the union bound and expect the term d to appear as a scaling factor, and so on). The method is elegant and Figure 1 supports it quite well (which shows that the true expectation is indeed sandwiched between the two terms claimed in the paper).

Clarity: The paper is very well-written and a pleasure to read.

Relation to Prior Work: The relation to prior work is clearly discussed.

Reproducibility: Yes

Additional Feedback: 1- It would an important addition if the authors show how the LP in (3) can be solved using variants of SGD. Otherwise, the current method cannot be applied to large amounts of data. In Table 1, all of the classification problems use less than 1,000 training examples. Please correct me if I am wrong about the number of training examples of each dataset. 2- There should be some discussion on what would be a good feature map. For example, I would expect a good set of m feature maps to be diverse (different from each other). Also, we would want the variance of the feature map to be as large as possible, depending on the available sample size, which puts a limit on variance as stated in Theorem 3. A few lines about this can be helpful such as at the Conclusion section. 3- Please mention the criteria used for choosing the six UCI datasets in Table 1.


Review 3

Summary and Contributions: This paper presents minimax risk classifiers (MRCs) that do not rely on a choice of surrogate loss and family of rules. The goal of MRC is to find a classification rule that minimize the worst-case expected 0-1 loss with respect to a class of possible distributions. It first represents data, probability distributions and classification rules by matrices. The estimated classifier is cast as a linear optimization problem in which the uncertainty set is cast as the linear constraints. Some performance guarantees are proved, and numerical comparisons are conducted. ---- update: I have read the rebuttal. It is claimed that "the methods presented do not create or compute matrices describing probability distributions and classification rules", but the formula at the end of page 3 clearly indicates that p is used to describe the probability mass function for all possible data point x (there \ell is the expected value of the misclassification loss (1-h)). Perhaps p is not needed for the training of the classifier; however, the formula at the bottom of page 3 should clearly be correctly presented as the empirical risk instead of the population risk and the p therein should be the empirical probability distribution. This is at least sloppy writing. Moreover, the feature maps have not been satisfactorily explored. Some illustrative examples and a discussion on how to choose Phi will help a lot. Overall I believe this work has a lot of merits but I believe a thorough revision will make it much better. I have bumped my rating by 1.

Strengths: There is some novelty in the proposed method. Theoretical guarantees are provided.

Weaknesses: 1. The proposed method is inapplicable to data from absolutely continuous probability distribution. The number of possible values of a data point in this case will be infinite. However, the paper relies on the vectorization of the probability distribution. For truly real world continuous data, huge matrices will have to be created and computed. 2. The choice of the uncertainty set is somewhat arbitrary. How to guarantee that the true distribution is covered? 3. There has to be a trade off between the data fitting and the generalization error. This seems to be related to how the uncertainty set is defined. In the extreme case that the uncertainty set is chosen to be infinitely large, then the model will underfit the data. Insight about this trade off is lacking. 4. The linear program in Theorem 3 need to be explained intuitively. I understand that this is a main theorem but it would help the reader a lot if the authors can explain what are the objective and the constraints in (3). 5. How is the feature mapping chosen? How sensitive is the model to the feature mapping? 6. I am not very much impressed by the numerical study. Cross-validation or other careful tuning methods should be used for the other SOTA methods to compare with the current method.

Correctness: Looks alright.

Clarity: Some explanation to the linear program (3) in Theorem will be good.

Relation to Prior Work: Yes.

Reproducibility: No

Additional Feedback:

[Author Response · NeurIPS 2020]

In the following, we respond to all the reviewers' questions that will be addressed in the paper's final version together with all their suggestions.

**Reviewer #1**

1. The optimization of 0-1 loss instead of a surrogate loss brings the learning process one step closer to the original goal of minimizing expected 0-1 loss. The results in the paper show that optimizing 0-1 loss can lead to enhanced performance guarantees (much tighter bounds) since the learning process directly provides bounds on the expected 0-1 loss (probability of error). We would also like to point out that the bounds' tightness is shown not only in Fig.1 but also in "LB" and "UB" columns of Table 1.

2. Learning techniques in Theorem 1 and performance guarantees in Theorems 2 and 3 are obtained addressing the minimax in line 105 using Lagrange duality. For instance, parameters $\boldsymbol{\mu}_a$ and $\boldsymbol{\mu}_b$ correspond to the Lagrange multipliers of constraints given by $\mathbf{a}$ and $\mathbf{b}$ in (1), respectively. (See also Answer 4 to Reviewer #4)

3. cvxpy can be installed following the instructions in https://www.cvxpy.org/install/. We will also include a readme file with detailed installation steps in the implementation files.

**Reviewer #2**

1. As the Reviewer mentions, optimization based on stochastic gradient descent (SGD) approaches can increase efficiency especially for large-scale training. The learning techniques proposed in the paper can be addressed using variants of SGD methods. In particular, primal-dual subgradient descent methods can enable efficient iterative optimization using subgradients of objective and constraints functions. In addition, the expression for $\mathbf{a}$ and $\mathbf{b}$ in (2) given by sample averages leads to an objective function in (3) that is amenable for stochastic subgradient descent methods.

2. All methods (proposed MRC and competing techniques) were implemented using their default parameters and settings in all datasets for a fair and transparent experimental comparison.

3. The role of the feature map in the presented methods is similar to that in conventional linear classifiers such as SVMs and logistic regression. However, as the Reviewer mentions, the paper offers new insights for feature mappings' design including their role in determining the probability distributions considered (uncertainty set) in (1) together with the trade-offs for dimensionality, variability, and training size in the generalization bounds of Theorem 3.

4. The main criteria for choosing the UCI datasets was to select frequently used datasets for binary and multi-class problems. Those with large number of samples were used for comparison with performance bounds in Fig.1 over one instantiation in terms of training size up to 10,000 samples, while the others (with less than 1000 samples) were used for comparison with both state-of-the-art techniques and performance bounds in Table 1 using 10-fold cross-validation.

**Reviewer #4**

1. We would like to clarify that the finite cardinality of the instance space does not lead to any practical limitation or computational burden. In the paper, instance spaces are taken to be finite only for technical convenience in the proof of Theorem 1. Infinite instance spaces would require to use heavier tools from variational analysis in such proof, but the corresponding MRC methods would not change. The methods presented do not create or compute matrices describing probability distributions and classification rules. Such methods obtain expectation estimates $(\mathbf{a}, \mathbf{b})$ using (2) and MRCs parameters $(\boldsymbol{\mu}_a, \boldsymbol{\mu}_b, \nu)$ through optimization problem (3) that has dimensionality and number of constraints given by the feature mapping used, independently of the size of the instance space.

2. Uncertainty sets are chosen to be determined by linear constraints in (1) so that Lagrange duality enables to obtain efficient learning techniques through Theorem 1. In addition, the uncertainty sets proposed can be guaranteed (with prob. $> 1 - \delta$) to include the true data-generating distribution using $(\mathbf{a}, \mathbf{b})$ given by expectations' confidence intervals at level $1 - \delta$. Such condition can be achieved by using parameter $\lambda$ as given in Theorem 3 (line 183) or using other statistical methods that obtain expectations' confidence intervals from i.i.d. samples.

3. MRCs' generalization depends on the uncertainty set that is determined by the feature map in (1). Theorem 3 shows MRCs' generalization in terms of feature map characteristics such as dimensionality and variability. We will describe how to interpret such results in terms of uncertainty sets, e.g., increased dimensionality reduces uncertainty set size.

4. We will include a short description of Theorem 1 proof to show the intuition behind such result. In particular, the minimax problem addressed by MRCs is equivalent to optimization problem (3) by using Lagrange duality and maximin equivalence. Parameters $\boldsymbol{\mu}_a, \boldsymbol{\mu}_b$, and $\nu$ are the Lagrange multipliers corresponding to the linear constraints defining the uncertainty set, and constraints in (3) come from the conjugate of the objective in the maximin problem.

5. We will describe that the role of the feature map in the presented methods is similar to that in conventional linear classifiers such as SVMs and logistic regression. In particular, MRCs are determined by a linear-affine combination of the feature map as shown in (4). Threshold-based features are used in experimentation section to plainly show the potential of the new approach. More sophisticated feature maps such as those given by kernel-based embeddings in conventional techniques can be analogously used (see short discussion in lines 236-239 and footnote 1).

6. For fair experimental comparison, we implemented all the methods (including the proposed MRCs) using their default settings and parameters in all the datasets. For the experimentation carried-out in the paper, we consider that it is more transparent not to use cross-validation or tuning in any method.

[Meta-Review · NeurIPS 2020]

This paper presents an interesting new perspective on the design of learning methods: the idea is to choose a classifier that minimizes the risk function uniformly over a family of distributions, constructed based on an iid data set, with the guarantee that (with high probability) the true data-generating distribution is contained in the family. This inherently supplies an upper bound on the risk of the chosen classifier. The family of distributions is generated by constraints on the expectation of a function Phi of (x,y), using data-dependent confidence bounds on its true expectation to set the constraints. Thus, the method is highly dependent on the choice of the function Phi. One significant concern noted by the reviewers is that the paper doesn't seem to explore this dependence in much depth, such as providing an array of illustrative examples and design principles for Phi, discussion of how choices of Phi for a given sample size may relate to notions of expressiveness and overfitting, or checking whether the technique can provide guarantees competitive with known results obtained by more traditional approaches (e.g., kernel methods, or ERM guarantees from uniform convergence). In other words, they do not flesh out this new theory by investigating its potential connections to, or advantages over, previous approaches in any concrete scenarios. As such, it seems hard to guess what advantages this approach might provide, other than simply providing a new perspective. In summary, the new approach is innovative and seems like a potentially promising new direction. However, the concrete implications and advantages of the new theory aren't argued very thoroughly or convincingly.